# Capturing the primordial *Kras* mutation initiating urethane carcinogenesis

Siqi Li[1], David M. MacAlpine[1] & Christopher M. Counter [1✉]

The environmental carcinogen urethane exhibits a profound specificity for pulmonary tumors driven by an oncogenic $Q_{61}L/R$ mutation in the gene *Kras*. Similarly, the frequency, isoform, position, and substitution of oncogenic RAS mutations are often unique to human cancers. To elucidate the principles underlying this RAS mutation tropism of urethane, we adapted an error-corrected, high-throughput sequencing approach to detect mutations in murine *Ras* genes at great sensitivity. This analysis not only captured the initiating *Kras* mutation days after urethane exposure, but revealed that the sequence specificity of urethane mutagenesis, coupled with transcription and isoform locus, to be major influences on the extreme tropism of this carcinogen.

[1] Department of Pharmacology and Cancer Biology, Duke University Medical Center, Durham, NC 27710, USA. ✉email: count004@mc.duke.edu

Exposure of mice to the environmental carcinogen urethane primarily induces tumors in one organ (lung) with a single driver mutation in only one of the three *Ras* genes (*Kras*), at one position ($Q_{61}$), with one substitution (L or R depending on the mouse strain)[1–5]. This extreme RAS mutation tropism is rather remarkable considering that oncogenic Ras is well known to cause a host of cancers in the mouse beyond the lungs[6]. Further, there are 57 possible point mutations generating 54 known or potentially oncogenic substitutions among the three *Ras* genes (*Hras*, *Nras*, and *Kras*) at the three major oncogenic positions ($G_{12}$, $G_{13}$, and $Q_{61}$) from a total of six possible oncogenic amino acids (V, D, C, S, R, A at $G_{12}/G_{13}$ and L, R, K, H, E, P at $Q_{61}$)[7]. Capturing the mutations arising immediately after urethane exposure in normal tissues in vivo would greatly inform the underlying mechanism by which this carcinogen induces one tumor type driven by an incredibly specific Ras mutation.

The challenge of detecting mutations early lies in the extremely low rate of urethane mutagenesis in vivo, which ranges from 5 to $100 \times 10^{-6}$ or 1 mutant per 0.1 to $2 \times 10^5$ templates[3,8]. This rate is significantly lower than the detection limit of conventional next generation sequencing (NGS)[9], which varies from 1 to $10 \times 10^{-3}$ or 1 mutant per 100 to 1000 templates. Various techniques have been developed to lower the error rate of NGS[10,11], but improvements are still limited by mutations arising during early PCR steps and data recovery efficiency[12]. The recently developed error-corrected, high-throughput maximum-depth sequencing (MDS) method overcomes these limitations, identifying ultra-rare ($1 \times 10^{-6}$, 1 mutant per $10^6$ templates) antibiotic-resistance mutations arising in bacteria populations[13]. We thus sought to adopt this assay for the much larger mammalian genome to capture the initiating *Kras* mutation induced by urethane in vivo to elucidate the principles underpinning the extreme selectivity of this carcinogenic process.

By adapting MDS for the mammalian genome, we capture the dominant initiating $Kras^{Q61L}$ mutation in the lungs of mice immediately following urethane exposure. Further, we show that the substitution and position tropism of urethane can be largely ascribed to the specificity of this carcinogen for C<u>AN</u>➔C<u>TN</u> mutations, which generates the oncogenic $Q_{61}L$ mutation in *Kras*. The same mutations were also captured in *Hras*, arguing that these mutations are not sufficient to induce tumorigenesis, which speaks to the isoform tropism of urethane. Last, in terms of tissue tropism, oncogenic Ras mutations were generally undetectable in other organs tested, which we suggest is linked to the transcriptional status of genes. Collectively, these finding indicate that RAS mutation tropism is a multifactorial process, which may inform similar RAS mutation patterns observed in human cancers.

## Results

**Adapting MDS to the mammalian genome**. A barrier to detecting initiating mutations in *Kras* at the time they occur in vivo after urethane exposure is that the mutation rate of this carcinogen is well below the detection limit of NGS. To overcome this limitation, we turned to the error-corrected, high-throughput sequencing approach of MDS, which recovered mutants in bacteria at a frequency as low as $1 \times 10^{-6}$ or 1 mutant per $10^6$ templates[13]. The key steps of MDS are first, synthesis of unique barcodes onto one strand of a genomic region-of-interest, second, linear amplification to obtain multiple direct copies of the barcoded genomic DNA, third, exponential amplification to obtain families of PCR products sharing the same barcode, and fourth, ultra-deep sequencing of millions of barcode families from the single region-of-interest[13]. Bona fide mutations are differentiated from PCR and sequencing errors by virtue of being detected in all members of one barcode family[13]. The challenge of adapting

MDS to the mammalian genome is maintaining the recovery of a sufficient number of analyzable barcode families (with at least two or three members) in a genome that is three orders of magnitude larger in size and weight[14,15]. To this end, we optimized assay conditions (see Methods) for mammalian *Kras* (Supplementary Fig. 1a) and barcode recovery (Supplementary Table 1). To validate the sensitivity of this mammalian version of MDS, we generated a panel of *Kras*-mutant plasmids, each comprised of *Kras* cDNA with a unique set of co-occurring double or triple mutations in the region encoded by exon 1 and/or exon 2 (Supplementary Table 2). Each was spiked at specific concentrations into genomic DNA isolated from mouse embryonic fibroblasts (MEFs) or murine lungs to benchmark different levels of sensitivity. As the error rates of PCR and sequencing are unlikely to give the same two or three exact improper base calls, the actual frequency of mutants present in the sample was estimated by calculating the frequency of barcode families with the pre-engineered co-occurring mutations. The frequency of mutations determined by MDS was then compared against the aforementioned actual frequency. Using this approach, we demonstrated that MDS adapted for the transcribed strand of *Kras* exon 1 detected mutations at a sensitivity of $5 \times 10^{-7}$ or 1 mutant per $2 \times 10^6$ templates (Fig. 1a, Supplementary Fig. 1b, and Supplementary Table 2). We further validated the sensitivity of the MDS assay adapted for the non-transcribed strand of *Kras* exon 2 in the same fashion (Supplementary Fig. 1c and Supplementary Table 2). Thus, MDS optimized for mammalian genomic DNA detects mutations at a sensitivity potentially 20,000 times greater than conventional NGS.

**Capturing the initiating oncogenic mutation in *Kras***. Urethane induces pulmonary tumors driven by a $Kras^{Q61L/R}$ oncogenic mutation[1,3–5], exemplifying the selectivity of this carcinogen at the level of tissue, isoform, position, and substitution. To elucidate the processes behind this RAS mutation tropism, we exposed A/J mice to urethane or the vehicle PBS via three daily intraperitoneal injections. After 1, 2, 3, and 4 weeks, genomic DNA was isolated from the lungs of four to seven mice from each condition. The non-transcribed strand of exon 2 of the endogenous *Kras* gene was then sequenced by MDS. To ensure abundant depth for mutant recovery and the accuracy of detected mutation frequency, samples with $<1.5 \times 10^5$ barcodes were excluded from analysis. For the remaining samples (barcode recovery listed in Supplementary Data 1), mutation frequencies were summed by either nucleotide position or substitution type, normalized to control PBS, $\log_{10}$ transformed, and then plotted as a heatmap (Fig. 1b, c). This analysis identified the well-established[1,3–5] oncogenic L (and to a lesser extent R) mutation at codon $Q_{61}$ preferentially in the urethane, but not PBS cohort of mice, as early as 1 week after exposure to this carcinogen. Consistent with being initiating events, these mutations expanded over time indicative of tumor growth (Fig. 1b and Supplementary Data 2), although a longer time course would formally confirm a tumorigenic identity. We also independently confirm by droplet digital PCR[16] the presence of the $Q_{61}L$ mutation 4 week post urethane exposure at a frequency similar to that determined by MDS (Supplementary Fig. 1d). We thus capture and confirm the primordial initiating oncogenic mutation in *Kras* within days of exposure to urethane.

**Substitution tropism**. Previous whole-exome sequencing of urethane-induced tumors revealed a strong bias toward A>T/G substitutions[3], consistent with ethenodeoxyadenosine adducts forming in vivo after urethane exposure[17,18]. These substitutions were also detected in *Kras* by MDS at a high frequency, although

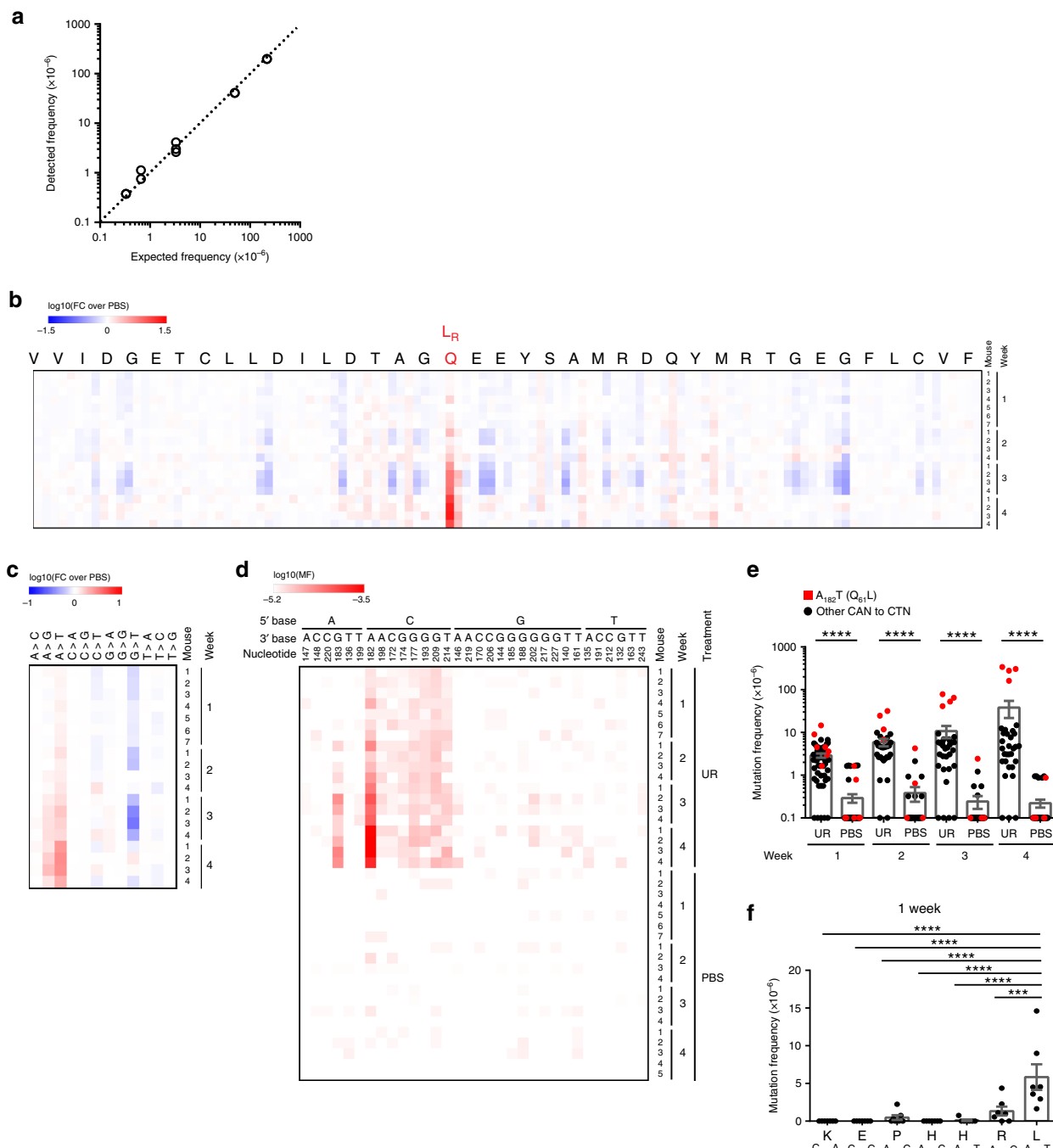

**Fig. 1 MDS detects ultra-rare mutations induced by the carcinogen urethane. a** Frequency of single (detected) versus co-occurring (present) mutations identified by MDS using a dilution series of *Kras* cDNAs with 2–3 different mutations engineered in exon 1 mixed with genomic DNA from mouse lung tissue. **b**–**d** Heatmap of the mutation frequency (MF) determined by MDS for the non-transcribed strand of exon 2 of *Kras* from the lungs of mice at the indicated time points after exposure to urethane (UR) or PBS ($n = 7$ mice for the UR and PBS cohorts at week 1, 5 mice for the PBS cohort at week 4, and 4 mice for all other cohorts from one experiment), plotted as (**b**, **c**) the log-transformed fold-change normalized to PBS-treated mice (FC over PBS) or (**d**) log-transformed versus each (**b**) nucleotide (annotated by amino acid at the top, $Q_{61}L$ and R mutations are highlighted in red, scaled by detected frequency), (**c**) type of substitution or (**d**) A>T transversions (nucleotide number as well as the 5′ and 3′ base of the substituted A are shown at the top). **e**, **f** Mean ± SEM mutation frequency of (**e**) each possible C$\underline{A}$N to C$\underline{T}$N transversion at the indicated time points after mice were exposed to urethane (UR) or PBS ($n = 7$ mice for the UR and PBS cohorts at week 1, 5 mice for the PBS cohort at week 4, and 4 mice for all other cohorts from one experiment) or (**f**) all possible missense mutations for $Q_{61}$ codon in mice 1 week after urethane exposure ($n = 7$ mice from one experiment). *p* values calculated by **e** Dunn's multiple comparison test following Kruskal–Wallis test or **f** Holm-Sidak multiple comparisons test following one-way ANOVA. ****$p < 0.0001$ and ***$p < 0.001$.

A>T transversions were far more common than A>G transitions (Fig. 1c). In agreement with this bias, $C\underline{A}_{182}A \rightarrow C\underline{T}A$ gives rise to the dominant $Q_{61}L$ oncogenic mutation in tumors of the A/J strain of mice exposed to urethane, while $C\underline{A}_{182}A \rightarrow C\underline{G}A$ gives rise to the rarer $Q_{61}R$ oncogenic mutation[1,5]. Still, the overall A/T content of murine genome[15] is about 58%, and A>T/G substitutions represents two-thirds of the possible base changes for this nucleotide. As such, this mutagenic signature is rather general compared with the extreme specificity of the initiating mutation. Further analysis of the mutation signature, using the $\log_{10}$-transformed mutation frequencies of individual substitutions, revealed that the most prominent substitution detected by MDS in the lungs of mice after urethane (but not PBS) exposure at all time points was an A>T transversion within the context of a 5′ C and 3′ any nucleotide, namely a C$\underline{A}$N trinucleotide (Fig. 1d, Supplementary Fig. 2a, b, and Supplementary Data 2, 3). In agreement, a 5′ C was favored to some extent for A>T transversions in previous whole-exome sequencing of urethane-induced lung tumors[3]. The frequency of C$\underline{A}$N$\rightarrow$C$\underline{T}$N mutations recovered in the urethane-exposed cohort remained constant over time in all but one case; $C\underline{A}_{182}A \rightarrow C\underline{T}A$ encoding the oncogenic $Q_{61}L$ mutation expanded at subsequent time points ostensibly due to tumor growth (Fig. 1e). The same was true for the second most prominent urethane-specific substitution detected by MDS, an A>G transition preceded by 5′ C (Supplementary Fig. 2b), where again $C\underline{A}_{182}A \rightarrow C\underline{G}A$ that gives rise the rarer $Q_{61}R$ oncogenic mutation[1,5] expanded over time (Supplementary Fig. 2c). Substitutions other than $C\underline{A}_{182}A \rightarrow C\underline{T}/GA$ at codon 61 were rarely detected 1 week after urethane exposure (Fig. 1f), even though all the possible missense mutations at this codon generated by a single-nucleotide substitution ($Q_{61}L$, R, K, E, P, and H) have been reported in human cancers in the COSMIC database[19]. As such, an A>T/G substitution preceded by C greatly increases the specificity of urethane mutations for codon 61, reducing the number of potential non-synonymous changes in both strands of the murine *Kras* gene by fivefold, from 616 to 120. The selectivity of these two substitutions after urethane exposure thus appears to be a major contributing factor to the substitution bias toward $Q_{61}L/R$ mutations in *Kras*.

**Position tropism.** This bias of urethane for (C)A>T/G substitutions similarly argues against mutations arising at an appreciable level in codons 12 ($G_{34}GT$) or 13 ($G_{37}GC$) in exon 1, as neither fit the CAN pattern in either strand orientation. Related to this, despite the fact that oncogenic mutations at $G_{12}$, and to a lesser extent $G_{13}$, occur frequently in human cancers[19] and when introduced into the lungs of mice are tumorigenic to varying degrees[20], they are rarely recovered from urethane-induced tumors[3]. We therefore sequenced the transcribed strand of exon 1 of *Kras* by MDS from genomic DNA isolated from the lungs of mice 1, 2, 3, and 4 weeks after exposure to urethane or PBS. To overcome interference from strand-specific background (see Methods), we also sequenced the non-transcribed strand of exon 1 of *Kras* by MDS from the lungs of mice at the 1- and 4-week time points. While C$\underline{A}$N$\rightarrow$C$\underline{T}$N transversions were again preferentially detected 1 week after urethane exposure (Fig. 2a, b, Supplementary Fig. 3a, b, and Supplementary Data 4, 5), indicating urethane mutagenesis occurred in this exon, oncogenic mutations were rarely recovered in either codon 12 or 13 (Fig. 2c). Interestingly, some $G_{12}$ and $G_{13}$ mutations were detected at a low frequency 4 weeks after urethane exposure (Fig. 2d). It is worth noting that oncogenic mutations at $G_{12}$ have been reported in urethane-induced tumors[21], but are quite rare. This suggests that $G_{12}$ and $G_{13}$ mutations are induced by urethane exposure, but remain below the limit of detection of MDS

unless a certain degree of clonal expansion occurs. Similarly, while a $Q_{61}H$ ($A_{183}$>T) mutation was rarely detected 1 week after urethane exposure (Fig. 1f), it was more prevalent in later samples (Fig. 1d and Supplementary Data 2). Collectively, these findings argue that the mutational position tropism of urethane can be ascribed in large part to a mutational bias of this environmental carcinogen toward C$\underline{A}$N$\rightarrow$C$\underline{T}$/G$\underline{N}$ mutations.

**Isoform tropism.** The other two *Ras* genes, *Hras* and *Nras*, encode the identical codon 61 (CAA). C$\underline{A}$N$\rightarrow$C$\underline{T}$/G$\underline{N}$ substitutions at this codon generate the identical oncogenic $Q_{61}L/R$ mutations, which are well known to render *Hras* and *Nras* oncogenic[22,23]. Despite this, oncogenic mutations in *Hras* or *Nras* are not recovered in urethane-induced lung tumors[3]. This suggests that either these loci are resistant in some manner to urethane mutagenesis or oncogenic mutations in these two genes are unable to initiate tumorigenesis. To differentiate between these two possibilities, we optimized the MDS assay to detect mutations in the non-transcribed strand of exon 2 in *Hras* (see Methods). We then applied this approach to genomic DNA isolated from the lungs of mice 1 and 4 weeks after exposure to urethane or PBS. We found a high prevalence of A>T followed by A>G mutations in exon 2 of *Hras* (Fig. 3a and Supplementary Data 6), with again C$\underline{A}$N$\rightarrow$C$\underline{T}$N transversions being the predominant mutation in the urethane cohort, including the oncogenic $C\underline{A}_{182}A \rightarrow C\underline{T}A$ mutation in codon 61 (Fig. 3b). C$\underline{A}$N$\rightarrow$C$\underline{T}$N transversions in *Hras* were detected somewhat less often than in *Kras* 1 week (Fig. 3c), but similarly 4 weeks after urethane exposure (Fig. 3d). Unlike in the case of *Kras*, however, oncogenic mutations in *Hras* did not expand appreciably over time (Fig. 3d). *Hras* therefore appears to acquire oncogenic mutations at a detectable frequency, but such mutations do not support tumorigenesis. This suggests that the isoform tropism of urethane is a product of the *Hras* locus and not an inability to induce oncogenic mutations at this site.

**Organ tropism.** Pulmonary lesions are the primary tumors arising in mice after intraperitoneal injections of urethane[2]. However, activating an oncogenic *Kras* allele in a broad spectrum of murine organs has been documented to be tumorigenic[24]. This begs the question of why urethane fails to induce other types of tumors. We thus analyzed the mutation status by MDS of the non-transcribed strand of exon 2 of *Kras* from lung compared with the liver and pancreas from mice 1 and 4 weeks after exposure to urethane versus PBS. The liver was chosen as in rare cases tumors develop in this organ during urethane carcinogenesis[25,26]. The pancreas was chosen as it is sensitive to tumorigenesis by oncogenic *Kras* mutations[27,28] but is not known to develop tumors after intraperitoneal injections of urethane[2]. In comparison with the lung, significantly fewer C$\underline{A}$N$\rightarrow$C$\underline{T}$N transversions were recovered in the liver and pancreas 1 and 4 weeks after urethane exposure (Fig. 4a, b and Supplementary Data 7, 8). Again, unlike the situation in the lung, there was no overt expansion of *Kras* oncogenic mutations in the liver and pancreas over time, suggesting an absence of tumor growth (Fig. 4b). These findings argue that *Kras* acquires fewer mutations in these tissues after urethane exposure. To rule out the possibility that these tissues are less exposed to urethane, mice were injected with urethane or PBS and 2, 4, and 8 hours later the lungs, liver, and pancreas were removed and subjected to LC/MS/MS[29] to measure the levels of urethane and its active metabolite vinyl carbamate[2,30]. Similar levels of both compounds were detected in the lungs and liver, but less in the pancreas over the three time points, with the terminal time point showing the highest concentration in the liver, followed by the lung, and then the

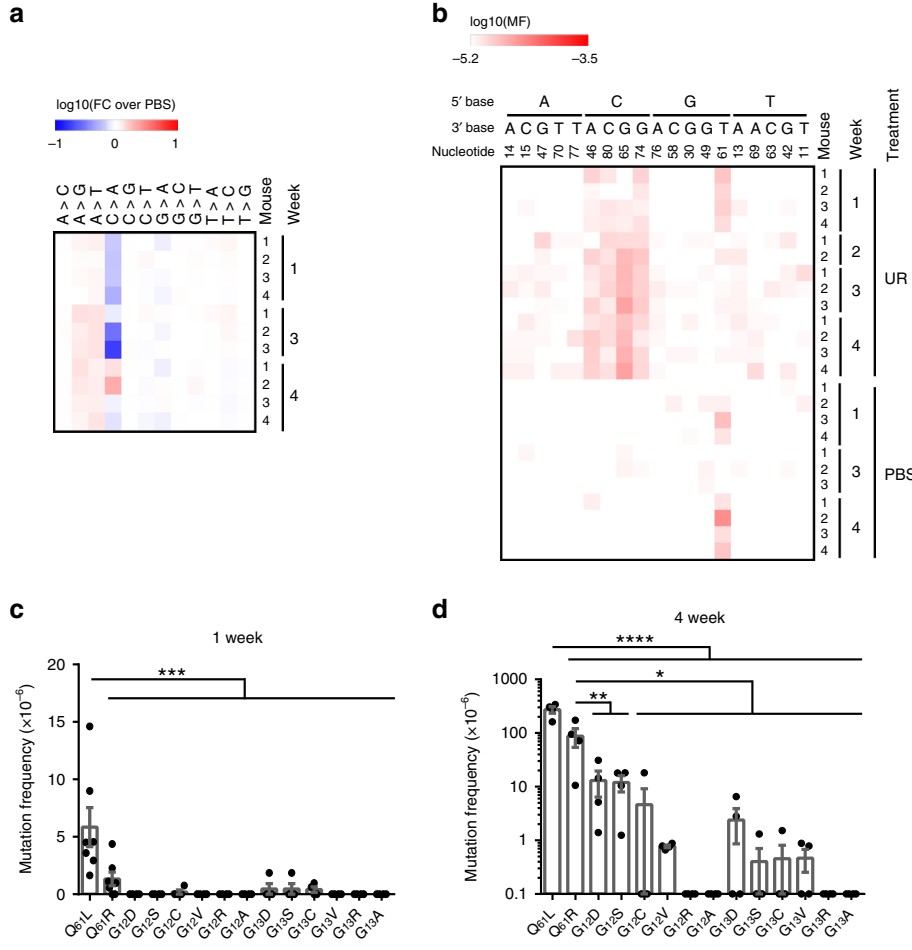

**Fig. 2 MDS detects the position and substitution tropism of urethane. a, b** Heatmap of the mutation frequency (MF) determined by MDS for the transcribed strand of exon 1 of *Kras* from the lungs of mice at the indicated time points after exposure to urethane (UR) or PBS (*n* = 2 mice for the UR cohort at week 2, 3 mice for the UR and PBS cohorts at week 3, and 4 mice for all other cohorts from one experiment), plotted as the (**a**) log-transformed fold-change normalized to PBS-treated mice (FC over PBS) or (**b**) log-transformed versus each (**a**) type of substitution or (**b**) A>T transversions (nucleotide number as well as the 5′ and 3′ base of the substituted A are shown at the top). **c, d** Mean ± SEM mutation frequency for all possible oncogenic mutations at $G_{12}$ and $G_{13}$ (*n* = 4 mice from one experiment) compared with the previous determined mutation frequency at codon $Q_{61}$ (Fig. 1, *n* = 7 mice at week 1 and 4 mice at week 4 from one experiment) (**c**) 1 week or (**d**) 4 weeks after urethane exposure. *p* values calculated by Holm-Sidak multiple comparisons test following one-way ANOVA. ****$p < 0.0001$, ***$p < 0.001$, **$p < 0.01$, and *$p < 0.05$.

pancreas (Supplementary Fig. 4a, b), similar to results from the lung and liver using radiolabeled urethane[31]. These findings argue that the organ tropism of urethane appears to arise from differences in mutagenesis between tissues, rather than differential carcinogen exposure.

**Strand bias**. Given the above differences in the mutation frequency between different tissues, we revisited the MDS sequencing of the *Kras* locus, finding a bias toward mutations in the non-transcribed strand in mice exposed to urethane. In more detail, MDS targeting the non-transcribed strand of *Kras* exon 2 revealed that CAN→CTN, but not the complement NTG→NAG transversions, were the predominant mutations in the lungs of mice 1 week after exposure to urethane (Fig. 5a). To independently validate this result, we performed MDS targeting the opposite (transcribed) strand of exon 2 of *Kras* from the lungs of mice 3 weeks after exposure to urethane or PBS (Supplementary Data 3). This revealed a bias toward NTG→NAG over CAN→CTN transversions in the transcribed strand (Fig. 5b). The same was true for exon 1 of *Kras*, namely a bias toward CAN→CTN transversions in the non-transcribed strand compared with the transcribed strand, as determined from sequencing

both strands by MDS (Supplementary Fig. 5a, b). Thus, based on sequencing both strands in two different exons of *Kras* in the lung, urethane mutagenesis exhibits a bias for the non-transcribed strand.

Mutational strand asymmetry has been observed for other mutational processes[32,33] and correlated with the transcriptional status of mutated genes[34]. *Kras* mRNA levels determined by quantitative RT-PCR (RT-qPCR)[35] or RNA-seq[36] have been reported to be higher in the murine lung compared with the liver. In agreement, we validated the higher expression of *Kras* mRNA in lung compared with liver and pancreas by RT-qPCR (Fig. 5c). In addition, strand bias is not significant in the liver, consistent with a general lack of mutations detected in this organ after urethane exposure (Fig. 5d). Prompted by this, we examined the relationship between mutation frequency and gene expression using mutations detected in a previous published whole-exome sequencing of urethane-induced lung adenomas and adenocarcinomas[3] and a published RNA-seq dataset generated from the adult mouse lung[36]. Genes were partitioned into quartiles based on expression level and the number of CAN→CTN transversions in non-transcribed or transcribed strand in each quartile determined. In agreement with the transcriptional strand bias

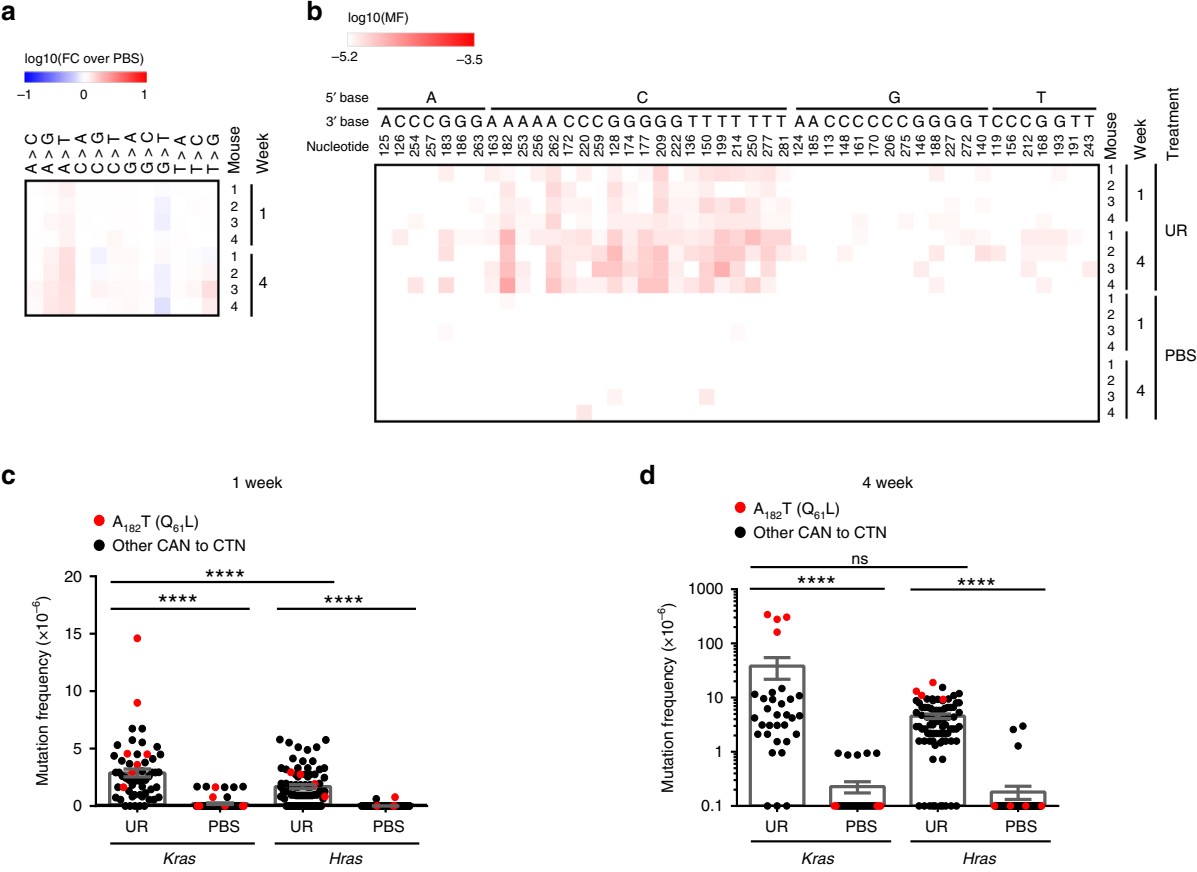

**Fig. 3 MDS detects the isoform tropism of urethane. a, b** Heatmap of the mutation frequency (MF) determined by MDS for the non-transcribed strand of exon 2 of *Hras* from the lungs of mice at the indicated time points after exposure to urethane (UR) or PBS ($n = 4$ mice at each time point from one experiment), plotted as the (**a**) log-transformed fold-change normalized to PBS-treated mice (FC over PBS) or (**b**) log-transformed versus each (**a**) type of substitution or (**b**) A>T transversions (nucleotide number as well as the 5′ and 3′ base of the substituted A are shown at the top). **c, d** Mean ± SEM mutation frequency of each possible CAN to CTN transversion in exon 2 of *Hras* ($n = 4$ mice from one experiment) compared with the previous determined mutation frequency in exon 2 of *Kras* (Fig. 1, $n = 7$ mice at week 1 and 4 mice at week 4 from one experiment) at (**c**) 1 week or (**d**) 4 weeks after exposure to urethane (UR) or PBS. *p* values calculated by **c** Holm-Sidak multiple comparisons test following one-way ANOVA, or **d** Dunn's multiple comparison test following Kruskal–Wallis test. ****$p < 0.0001$ and ns: not significant.

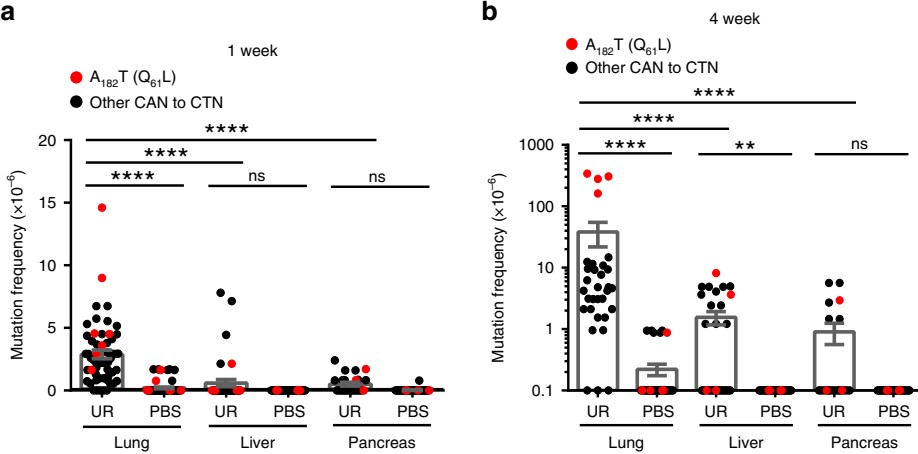

**Fig. 4 MDS detects the organ tropism of urethane. a, b** Mean ± SEM mutation frequency each possible CAN to CTN transversion determined by MDS for the non-transcribed strand of exon 2 of *Kras* in the pancreas or liver of mice ($n = 3$ mice for pancreas samples from the UR cohort at weeks 1 and 4, 5 mice for liver samples from the UR and PBS cohorts at week 1, and 4 mice for all other cohorts from one experiment) compared with the previous determined mutation frequency in the lung (Fig. 1, $n = 7$ mice at week 1 and 4 mice at week 4 from one experiment) either (**a**) 1 week or (**b**) 4 weeks after exposure to urethane (UR) or PBS. *p* values calculated by **a** Holm-Sidak multiple comparisons test following one-way ANOVA or **b** Dunn's multiple comparison test following Kruskal–Wallis test. ****$p < 0.0001$, **$p < 0.01$, and ns: not significant.

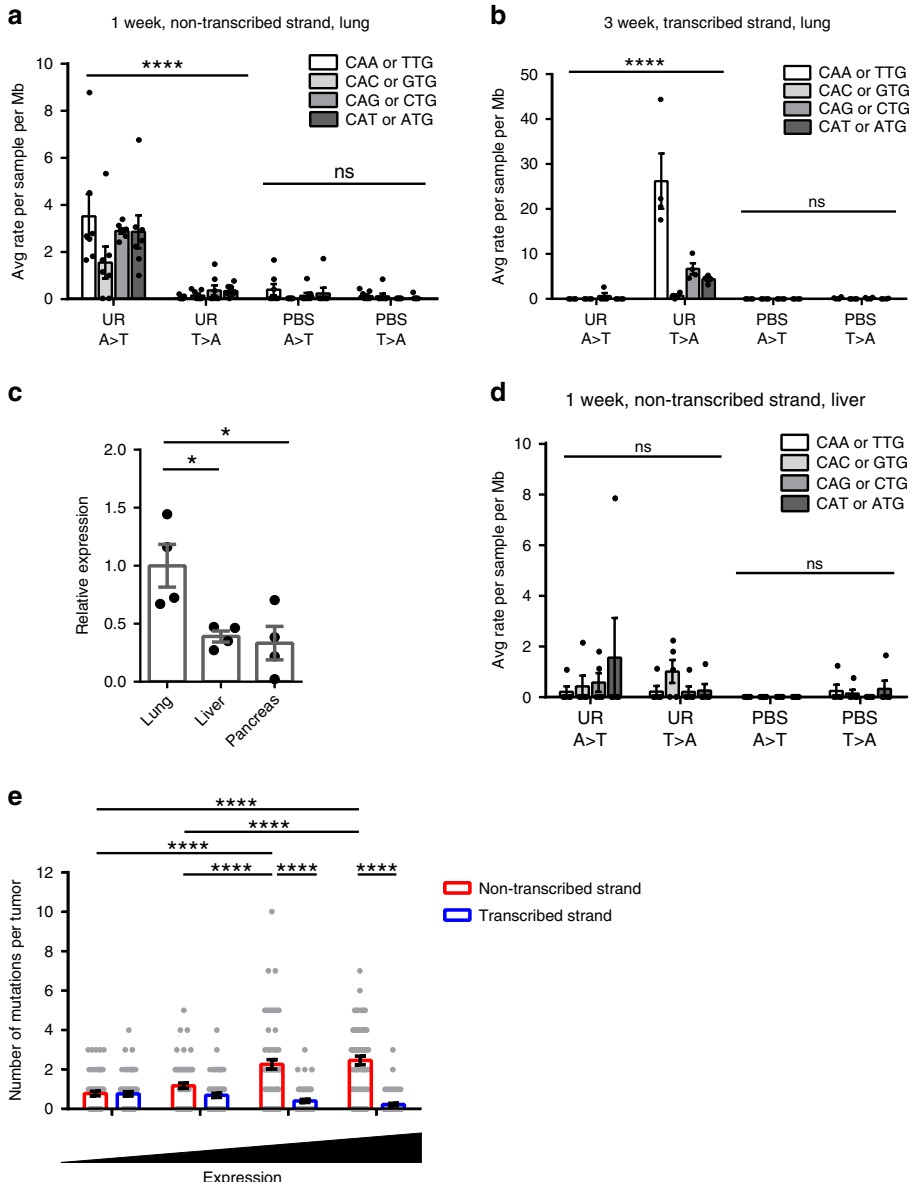

**Fig. 5 Urethane strand bias. a, b, d** Mean ± SEM mutation frequency of the indicated CAN to CTN transversions and reverse-complementary substitutions averaged by nucleotide positions determined by MDS sequencing of *Kras* exon 2 (**a**) non-transcribed strand in lungs (Fig. 1, $n = 7$ mice from one experiment) or (**d**) non-transcribed strand in livers of mice 1 week ($n = 5$ mice from one experiment) or (**b**) transcribed strand in lungs of mice 3 weeks ($n = 4$ mice from one experiment) after exposure to urethane (UR) or PBS. **c** Mean ± SEM relative expression of Kras mRNA in the lung, liver, and pancreas (normalized to lung) of mice determined by RT-qPCR ($n = 4$ mice from one experiment). **e** Mean ± SEM frequency of CAN to CTN transversions of the non-transcribed versus transcribed strand in urethane-induced tumors from whole-exome sequencing data[3] in genes binned by their mRNA levels from the mouse lung[36] ($n = 66$ tumors). *p* values calculated by Holm-Sidak multiple comparisons test following **a, b, d, e** two-way ANOVA or **c** one-way ANOVA. ****$p < 0.0001$, *$p < 0.05$ and ns: not significant.

revealed by MDS sequencing, CAN➔CTN transversions increased with gene expression on the non-transcribed strand but decreased on the transcribed strand (Fig. 5e). The same trends were observed when the RNA-seq dataset for adult mouse lung from the mouse ENCODE project[37] was analyzed (Supplementary Fig. 5c). Collectively, these findings point toward the organ tropism of urethane being related to the high transcription of *Kras* in the lung.

## Discussion

Here we adapted MDS, an error-corrected, high-throughput sequencing approach originally developed for use in microbiology[13],

to now detect extremely rare mutations in the mammalian genome at a sensitivity of up to $5 \times 10^{-7}$ (1 mutant per $2 \times 10^6$ templates). While we developed this assay to study RAS mutation tropism, MDS could find value in other applications[13], such as early detection[38]. Nevertheless, by leveraging MDS to study the mutagenesis process at the earliest stage of tumorigenesis, we detected the initiating $Q_{61}L/R$ mutations in *Kras* in the lungs of mice only days after exposure to urethane, capturing the very birth of cancer. We note that mutant allele-specific amplification[39,40] and droplet digital PCR[41] have documented *Kras* mutations after carcinogen exposure. However, we chose to develop MDS for the mammalian settings as these assays are either not as quantitative and sensitive[39,42], or are designed to examine pre-selected mutations[41,43]. Indeed, capitalizing on the

ability of MDS to detect potentially any sequence variation in targeted regions of *Ras* genes at great sensitivity, we show at least three features underpinning the extreme mutational tropism of urethane—the mutational bias of this environmental carcinogen, transcription, and the gene locus.

With regards to the substitution and position bias of urethane, we demonstrate that the prevalence of $Q_{61}L/R$ mutations arises in large part due to the known preference of urethane for A>T/G substitutions[3], especially as we show here in the context of a 5′ C. This mutational bias, coupled with codon 61 containing a CAN trinucleotide that when the A is mutated to either T or G gives rise to an oncogenic L (CT$_{182}$A) or R (CG$_{182}$A) amino acid, favors the $Kras^{Q61L/R}$ driver mutation characteristic of this carcinogen. Other oncogenic mutations at $Q_{61}$, $G_{12}$, or $G_{13}$ codons do not result from CA→CT/G substitutions, and in agreement, were rarely detected following urethane exposure. The implication being that a mutagenic preference may influence the type of initiating mutations in cancer. Similarly in humans, a CCT→CTC mutation characteristic of C>T transitions induced by UV encodes an activating $P_{29}S$ mutation in *RAC1* in sun-exposed melanoma[44].

While $Q_{61}H$, $G_{12}$, and $G_{13}$ oncogenic mutations in *Kras*, which are not favored by urethane mutagenesis, were rare or absent 1 week after urethane exposure, they were detected 4 weeks later. This implies that extremely rare mutations induced by urethane, provided they have a favorable oncogenic outcome, may initiate tumorigenesis (although we cannot formally rule out that these were pre-existing mutations unveiled by a cooperating mutation induced by urethane). In agreement, while the $Q_{61}L$ mutation is more frequent than $Q_{61}R$ in urethane-induced lung tumors of the A/J mouse strain, the reverse is true in the B6 strain[5]. Similarly, the mutation spectrum of urethane is also shifted in a variety of mutant Ras backgrounds[3,24,45,46]. If the mutagenesis preference of urethane is independent of strain background, the prevalence of the $Q_{61}R$ mutation suggest that this less common mutation is more conducive to tumor initiation in the B6 strain. As such, the most dominant mutation of a mutagen may not always dictate the initiating event, echoing reported discordances between the mutagenic signatures and the putative initiating mutation in certain human cancers[47–49].

Another fascinating feature of urethane mutagenesis revealed by MDS sequencing relates to isoform tropism. We found that codon 61 was readily mutated in *Hras* in lung tissue, yet the oncogenic *Hras* allele was not expanded appreciably over time. This suggests that either Hras$^{Q61L}$ is not as oncogenic as Kras$^{Q61L}$ or the encoded protein is expressed too low (or high) to be tumorigenic. In support of the first, RAS isoforms differ in their residency at different membranes[50] and the composition of proteins within the immediate vicinity differs between RAS isoforms[51,52], with proteins like PIP5K1A[52], calmodulin[53], galectin-3[54], and so forth documented to specifically associate with KRAS. In support of the second, a *Kras* allele whereby the 3′ end was replaced with *Hras* exons to encode Hras protein was found mutated in urethane-induced tumors[55], indicating that under a *Kras* promoter Hras$^{Q61L}$ is indeed oncogenic in the lung. Whether the inability of oncogenic mutations in *Hras* to promote lung tumorigenesis is because the protein is less oncogenic, expressed too low, too high, combinations thereof, or for other reasons[7,24,56,57] remains to be elucidated. Nevertheless, the finding that *Hras* is mutated yet such mutations are not recovered in lung tumors[3] after urethane exposure is in itself an important finding, and perhaps related, of the three *RAS* genes, *HRAS* is mutated the least often in human cancers[6,7,58].

With regards to organ tropism, a very different mechanism appears to be at play. In this case, we found that *Kras* is rarely mutated in the liver and pancreas, despite the presence of the

carcinogen. While a number of factors could contribute to this variation in mutagenesis[59–61], one notable difference is that *Kras* mRNA levels are higher in the lung compared with these other tissues, suggestive of increased transcription. In fact, the lung was found to have the second highest levels of *Kras* mRNA of 15 adult murine tissues analyzed, second only to the brain[35]. *Kras* expression in the mouse lung also correlates with strain susceptibility to urethane carcinogenesis[62,63]. Related, we discovered that the non-transcribed strand of *Kras* is preferentially mutated, which for other mutagens has been linked to transcription-coupled repair of the transcribed strand[64] or transcription-coupled damage of the displaced, non-transcribed strand[34]. Indeed, we found a global correlation between mRNA levels and the mutation frequency of urethane. This is not to say that there is a universal concordance between high gene transcription and an elevated mutation frequency of the non-transcribed strand. Indeed, high transcription has been associated with a lower mutation frequency in chromatin-dense genomic regions in cutaneous squamous cell carcinomas[65]. Thus, the type of cancer, mutational process, specific genes, and so forth may influence the bias of a mutagenic process. In the case of urethane, however, we suggest that the tissue tropism is related to the high transcription of *Kras* in the lung, increasing the susceptibility of this gene to urethane mutagenesis.

In humans, there are also very distinct patterns to RAS mutations at the level of the organ (e.g., *RAS* is commonly mutated in pancreatic but rarely in breast cancer), isoform (e.g., *KRAS* is mutated in lung cancer while *NRAS* is mutated in melanoma), position (e.g., $G_{12}$ is mutated in CMML while $Q_{61}$ is mutated in thyroid carcinoma), and substitution (e.g., $G_{12}V$ is the primary mutation in bladder carcinoma while it is $G_{12}S$ in mouth carcinoma). There is no definitive mechanism to explain this phenomenon, although the pattern itself has been widely reported for decades[6,7,24,58,66–68]. In this regard, the extreme specificity of urethane carcinogenesis for $Kras^{Q61L/R}$-mutant pulmonary tumors may inform the basic principles of the RAS mutation patterns observed in these clinical samples. Admittedly, urethane is not a major environmental carcinogen in humans compared with, for example, tobacco smoke. $Kras^{Q61L/R}$ mutations are also rare in human lung cancers[24]. With these two provisos, we speculate that the RAS mutation tropism of human cancers may similarly be a product of mutagenesis selectivity factors, for example the specificity of the mutagenic process or susceptibility of a specific locus to mutations, and selection factors, for example differences in the oncogenic activity of one isoform over another. Moreover, it is entirely possible, if not likely, that different combinations of these or even other factors such as cooperating mutations, as elegantly demonstrated in MNU carcinogenesis[3], cell type[69], signaling intensity[45], and so forth[24] underlie the RAS mutation tropism human cancers. As such, each cancer initiating event may be molded by a unique set of factors, each with varying influence.

## Methods

**Cell culture**. Mouse embryonic fibroblasts (MEFs) derived from E13.5 mouse embryos were stably infected with an ecotropic retrovirus derived from pBabe-Hygro[70] encoding the early region of SV40[71] and selected with 100 µg.ml−1 hygromycin to establish immortalized cultures using standard procedures and then cultured in Dulbecco's modified Eagle's medium (DMEM) supplemented with 10% fetal bovine serum (FBS) and 1% penicillin-streptomycin.

**Construction of *Kras*-mutant plasmids**. A region upstream of *Kras* start codon was amplified from murine genomic DNA (termed PCR1). PCR reactions were comprised of 100 ng of genomic DNA, 2.5 µl of 10 µM forward (5′-AATTGCGG CCGCCCAGGGGGGTATAGCGTACTATGCAGAAT-3′) and reverse (5′-CATTT TCAGCAGGCCTTACAAT-3′) primers, 4 µl of 2.5 mM dNTP, 10 µl of 5X buffer (NEB), and 0.5 µl Q5® Hot Start High-Fidelity DNA Polymerase (NEB) in a total volume of 50 µl. PCR cycles were as follows: one cycle at 98 °C for 30 s, 28 cycles at

98 °C for 8 s, 64 °C for 15 s, 72 °C for 10 s, and one cycle at 72 °C for 2 min. PCR products were gel purified using QIAquick Gel Extraction Kit following the manufacturer's protocol (Qiagen).

Mutations in *Kras* cDNA were generated through error-prone PCR (termed PCR2). PCR reactions were comprised of 15 nmol of plasmid containing *Kras* cDNA, 2 µl of 10 µM forward (5′-ATTGTAAGGCCTGCTGAAAATGACTGAGT ATAAACTTGTGGT-3′) and reverse (5′-CAGGGTCGACTCACATAACTGTACA CCTTGTC-3′) primers, 2 µl of 2.5 mM dNTP, 1.25 µl of 50 mM MgCl₂, 2.5 µl of 10x buffer (Invitrogen), 5 µl of 2.5 mM MnCl₂, and 0.2 µl of Platinum Taq DNA polymerase (Invitrogen) in a total volume of 25 µl. PCR cycles were as follows: one cycle at 94 °C for 1 min, 18 cycles at 94 °C for 30 s, 55 °C for 30 s, 72 °C for 3 min, and one cycle at 72 °C for 3 min. PCR products were gel purified as described above.

Products from PCR1 and PCR2 were fused through overlap PCR (termed PCR3). Twenty nanograms product from PCR1 and 40 ng product from PCR2 were mixed with 4 µl of 2.5 mM dNTP, 10 µl of 5X buffer (NEB), and 0.5 µl Q5® Hot Start High-Fidelity DNA Polymerase (NEB) in a total volume of 50 µl reaction. PCR cycles were 98 °C for 30 s and 10 cycles at 98 °C for 8 s, 63 °C for 15 s, and 72 °C for 15 s. 2.5 µl of forward primer from PCR1 and 2.5 µl of reverse primer from PCR2 were then added and the reaction was continued in the following conditions: 98 °C for 30 s, 25 cycles at 98 °C for 8 s, 72 °C for 40 s, and one cycle at 72 °C for 2 min. PCR products were gel purified as described above.

Plasmid backbone was amplified from the pUC19[72] (Addgene 50005) plasmid (termed PCR4). PCR reactions were comprised of 1 ng of pUC19 DNA, 2.5 µl of 10 µM forward (5′-AATTGTCGACTTAGACGTCAGGTGGCAC-3′) and reverse (5′-TTAAGCGGCCGCGTTTGCGTATTGGGCGCT-3′) primers, 4 µl of 2.5 mM dNTP, 10 µl of 5X buffer (NEB), and 0.5 µl Q5® Hot Start High-Fidelity DNA Polymerase (NEB) in a total volume of 50 µl. PCR cycles were as follows: one cycle at 98 °C for 30 s, 28 cycles at 98 °C for 8 s, 65 °C for 15 s, 72 °C for 1 min, and one cycle at 72 °C for 2 min. PCR products were gel purified as described above.

Products from PCR3 and PCR4 were digested with SalI and NotI according the manufacture's protocol (NEB). Digested products were column purified using QIAquick PCR Purification Kit following the manufacturer's protocol (Qiagen), ligated, and transformed using standard methodologies. DNA was isolated from individual clones by NucleoSpin® Plasmid miniprep kit (MACHEREY-NAGEL) and validated by Sanger sequencing. Ten clones with different sets of co-occurring mutations in *Kras* exon 1 and/or 2 were selected to be spiked into wild-type mouse genomic DNA at different ratios to test the detection limit of maximum-depth sequencing (see below).

**Urethane treatment**. Six- to eight-week-old male and female A/J mice (JAX Stock #000646) were intraperitoneally injected daily for 3 days with either urethane (Sigma U2500) dissolved in PBS (1 g.kg⁻¹) or the vehicle PBS alone. Mice were humanely euthanized 1, 2, 3, or 4 weeks after the last injection and the lung, liver, and pancreas collected for the extraction of genomic DNA. All mouse care and experiments were performed in accordance with protocols approved by the IACUC of Duke University.

**Pharmacokinetic analysis**. Six- to eight-week-old male and female A/J mice (JAX Stock #000646) were intraperitoneally injected with one dose of urethane dissolved in PBS (1 g.kg⁻¹). Mice were humanely euthanized 2, 4, and 8 h later after which plasma, lungs, pancreas, and livers were harvested and snap frozen. Liquid chromatography (LC) tandem-mass spectrometry (MS/MS) was used to measure urethane (ethyl carbamate, EC) (Sigma U2500) and vinyl carbamate (VC) (Santa Cruz Biotechnology sc-213157) concentrations in plasma and tissues. The LC-MS/MS system consisted of Shimadzu 20A series LC and Applied Biosystems/SCIEX API 4000 QTrap MS/MS instrument. LC columns: Phenomenex C₁₈ 3 × 4 mm guard column (#AJ0-4287) and Agilent ZORBAX Eclipse Plus C₁₈ 150 × 4.6 mm 1.8-µm analytical column (#959994-902). Mobile phase A: 0.1% formic acid, 10 µM sodium acetate, and 2% acetonitrile; mobile phase B: 100% methanol. Elution gradient: isocratic flow 30% A. Flow rate: 0.8 ml.min⁻¹. The run time was 10 min. Calibration samples were prepared by adding pure standards of EC or VC to corresponding matrix (plasma or tissue) in appropriate concentration range. The calibration samples were analyzed alongside study samples as a single analytical batch on the day of analysis.

In 2 ml screw cap vial, 20 µl (EC) or 50 µl (VC) of plasma or tissue homogenate (1 part tissue and 2 parts water) diluted with water 1/100 (EC) or undiluted (VC), 10 µl of 2 µg.ml⁻¹ MC-d5 in water (internal standard) (Toronto Research Chemicals), and 60 µl (EC) or 100 µl (VC) of 20 mM xanthydrol (Sigma) in glacial HAc were added and incubated at room temperature for 30 min. Hundred microliters of water and 500 µl of chloroform were then added and the mixture was vigorously agitated (speed 4, 40 s; Fast-Prep FP120, Thermo Savant). After centrifugation at 16,000 × *g* for 5 min at room temperature, 200 µl (EC) or 400 µl (VC) of chloroform (lower) layer was subjected to a gentle stream of nitrogen for 30 min, dry residue reconstituted with 50 µl (EC) or 100 µl (VC) 50% A/50% B, centrifuged at 16,000 × *g* for 5 min at 4 °C, after which 5 µl (EC) or 10 µl (VC) was injected into LC-MS/MS system. The mass spectrometer was operated in positive mode with the following MRM transitions (*m/z*): 292/180.8 [EC-1st], 292/151.3 [EC-2nd], 297/181.8 [EC-d5-1st], 297/151.7 [EC-d5-2nd] for EC and 290/180.5

[VC-1st], 290/151.2 [VC-2nd], 297/181.8 [EC-d5-1st], 297/151.7 [EC-d5-2nd] for VC.

**Isolation of genomic DNA**. MEF cells were resuspended in lysis buffer (100 mM NaCl, 10 mM Tris pH 7.6, 25 mM EDTA pH 8.0, and 0.5% SDS in H₂O, supplemented with 20 µg.ml⁻¹ RNase A (Sigma)). Lung, pancreas, and liver (right lobe) from A/J mice (JAX Stock #000646) were cut into fine pieces and similarly resuspended in lysis buffer. Samples were incubated at 37 °C for 1 hr. Two microliters of 800 U.ml⁻¹ proteinase K (NEB) was then added to each sample, the samples were vortexed, and then incubated at 55 °C overnight. Genomic DNA was isolated by phenol/chloroform extraction followed by ethanol precipitation using standard procedures and quantified using Qubit fluorometer.

**Maximum-depth sequencing (MDS)**. The MDS assay[13] was adapted for mammalian *Ras* genes as follows. Twenty to fifty micrograms of genomic DNA was incubated with StuI (NEB) for analysis of the transcribed strand of *Kras* exon 1, EcoRV (NEB) and EcoRI (NEB) for analysis of the non-transcribed strand of *Kras* exon 1, XmnI (NEB) for analysis of the non-transcribed strand of *Kras* exon 2, and PleI (NEB) for analysis of the transcribed strand of *Kras* exon 2, or HphI (NEB) for the analysis of the non-transcribed strand of *Hras* exon 2. Reaction conditions were 5 units of the indicated restriction enzyme and per 1 µg DNA per 20 µl reaction (e.g., 20 µg genomic DNA, 5 µl enzyme (20 units.µl⁻¹), and 40 µl 10X buffer in 400-µl reaction). Digested genomic DNA was column purified using QIAquick PCR Purification Kit following the manufacturer's protocol (Qiagen) and resuspended in ddH₂O (35 µl H₂O per 10 µg DNA). The barcode and adaptor were added to the target DNA by incubating purified DNA with the appropriate barcode primer (see below) for one cycle of PCR. PCR reactions were comprised of 10 µg DNA, 2.5 µl of 10 µM barcode primer, 4 µl of 2.5 mM dNTP, 10 µl of 5X buffer (NEB), and 0.5 µl Q5® Hot Start High-Fidelity DNA Polymerase (NEB) in a total volume of 50 µl. The number of PCR reactions was scaled according to the amount of DNA. PCR conditions were 98 °C for 1 min, barcode primer annealing temperate (see below) for 15 s, and 72 °C for 1 min. One microliter of 20,000 U.ml⁻¹ exonuclease I (NEB) and 5 µl of 10X exonuclease I buffer (NEB) was then added to each 50 µl reaction to remove unused barcoded primers and incubated at 37 °C for 1 h and then 80 °C for 20 min. Processed DNA were column purified using QIAquick PCR Purification Kit as above and resuspended in ddH₂O (35 µl H₂O per column). The concentration of purified product was measured with Simpli-Nano spectrophotometer (GE Healthcare Life Sciences). Samples were linear amplified with forward adaptor primer (see below). PCR reactions were comprised of 1.5 µg DNA, 2.5 µl of 10 µM forward-adaptor primer, 4 µl of 2.5 mM dNTP, 10 µl of 5X buffer (NEB), and 0.5 µl Q5® Hot Start High-Fidelity DNA Polymerase (NEB) in a total volume of 50 µl. The number of PCR reactions was scaled according to the amount of DNA. PCR conditions were as follows: 12 cycles of 98 °C for 15 s, 70 °C for 15 s, 72 °C for 8 s. 2.5 µl of 10 µM exon-specific reverse primer (see below) and 2.5 µl of 10 µM reverse-adaptor primer (see below) were then added to each 50 µl reaction. The mixtures were then subjected to 20 cycles of exponential amplification. PCR conditions were as follows: 4 cycles of 98 °C for 15 s, exon-specific reverse primer annealing temperature (see below) for 15 s, 72 °C for 8 s, 16 cycles of 98 °C for 15 s, 70 °C for 15 s, and 72 °C for 8 s. The final library was size selected and purified with Ampure XP beads according to the manufacturer's protocol (Beckman Coulter). Sequencing was performed using HiSeq 2500 100 bp PE rapid run, HiSeq 4000 150 bp PE or NovaSeq 6000 S Prime 150 bp PE at Duke Center for Genomic and Computational Biology. For the optimization of barcode recovery, the same amount of genomic DNA was processed in parallel by MDS assay targeting *Kras* exon 1 transcribed strand and the PCR products were pooled together at different concentrations in one library to obtain different sequencing depths.

**Primers for maximum-depth sequencing**. Barcode primer: [Forward adaptor] [Index][Barcode][Primer]

Where

[Forward adpator] = 5′-TACGGCGACCACCGAGATCTACACTCTTTCCCT ACACGACGCTCTTCCGATCT-3′

[Index] = variable length of known sequences from 0 to 7 nucleotides (Supplementary Data 1)

[Barcode] = NNNNNNNNNNNNNNN

*Kras* exon 1 StuI [Primer] = 5′-CCTGCTGAAAATGACTGAG-3′ (annealing temperature: 60 °C)

*Kras* exon 1 EcoRV [Primer] = 5′-ATCTTTTTCAAAGCGGCTGGCT-3′ (annealing temperature: 68 °C)

*Kras* exon 2 XmnI [Primer] = 5′-TCTTCAAATGATTTAGTATTATTTATGG C-3′ (annealing temperature: 59 °C)

*Kras* exon 2 PleI [Primer] = 5′-TCAGGACTCCTACAGGAAAC-3′ (annealing temperature: 63 °C)

*Hras* exon 2 HphI [Primer] = 5′-TAGGTGGCTCACCTGTACTG-3′ (annealing temperature: 66 °C)

Forward-adaptor primer: 5′-AATGATACGGCGACCACCGAGAT-3′ (annealing temperature: 70 °C)

Exon-specific reverse primer: [Reverse adaptor][Index][Primer]

Where

[Reverse adaptor] = 5′-AAGCAGAAGACGGCATACGAGATCGGTCTCGGC
ATTCCTGCTGAACCGCTCTTCCGATCT-3′

(for barcode recovery optimization and *Kras* exon 2 mutant plasmid spike-in experiment) or

5′-CAAGCAGAAGACGGCATACGAGATGTGACTGGAGTTCA
GACGTGTGCTCTTCCGATCT-3′

(for all the other experiments)

[Index] = variable length of known sequences from 0 to 7 nucleotides (Supplementary Data 1)

*Kras* exon 1 StuI [Primer] = 5′-CTCTATCGTAGGGTCATACTCAT-3′ (annealing temperature: 62 °C)

*Kras* exon 1 EcoRV [Primer] = 5′-TATTATTTTTATTGTAAGGCCTGCTG
A-3′ (annealing temperature: 62 °C)

*Kras* exon 2 XmnI [Primer] = 5′-GACTCCTACAGGAAACAAGT-3′ (annealing temperature: 61 °C)

*Kras* exon 2 PleI [Primer] = 5′-CTTTCTTATTCAACTTAAACCCAC-3′ (annealing temperature: 59 °C)

*Hras* exon 2 HphI [Primer] = 5′-CTAAGCCGTGTTGTTTTGCAG-3′ (annealing temperature: 65 °C)

Reverse-adaptor primer: 5′-CAAGCAGAAGACGGCATACGAGA-3′ (annealing temperature: 70 °C)

All primers were synthesized by Integrated DNA Technologies (IDT).

**Data analysis**. All sequencing data were analyzed through the Galaxy web platform[73]. Specifically, raw data were uploaded to usegalaxy.org or Galaxy Cloudman. For analysis of *Kras* exon 1 transcribed strand in mouse lung tissue, only read 1 was used. For all the other experiments, read 1 and read 2 were joined via PEAR pairend read merger[74]. The reads were then filtered by quality by requiring 90% of bases in the sequence to have a quality core ≥20. Filtered reads were split into different files based on assigned sample indexes and variation in sequence lengths using the tool Barcode Splitter and the tool Filter Sequences by Length.

For the experiment optimizing barcode recovery, the reads were trimmed down to the barcode and grouped into families by barcode. The number of families containing 1 read and ≥2 reads was then counted, respectively.

For the mutant plasmid spike-in experiments, the reads were trimmed down to the barcode and the bases containing engineered mutations. Trimmed reads were grouped by barcode into different families. The frequency of mutants present was calculated by dividing the counts of families containing engineered co-occurring mutations by the total number of families. The frequency of mutants detected was calculated by dividing the counts of families containing ≥2 reads and have ≥90% reads sharing the same engineered mutation at one specified position by the total number of families.

For the experiments examining carcinogen-induced mutations in *Kras* exon 1 or 2, the reads were trimmed down to the barcode and the target exon. Trimmed reads were grouped by barcode. Barcode families containing ≥3 reads and a unique consensus sequence were selected. To ensure sufficient barcode recovery for the purpose of sensitivity and accuracy, samples with <1.5 × 10⁵ barcode families recovered were excluded from downstream analysis. Sequences from selected barcode families were compared against annotated reference mutant sequences containing all possible single-nucleotide substitutions in the exon of interest and the mutation in the reference mutant sequence was assigned to the matched barcode family. The frequency of the corresponding mutation was calculated by dividing the counts of the families containing the mutation by the total number of families.

C>T and G>T substitutions have high background in PBS-treated mouse and have been previously identified as artifacts caused by deamination of cytosine or methyl-cytosine or oxidation of guanine arising during in library preparation[11,75], or mis-incorporated nucleotides in vivo not yet repaired[13]. Consistent with this, we detect high C>T or G>T substitutions but not the complementary G>A or C>A substitutions from the strand processed by MDS (Supplementary Data 2–8). To circumvent this background, the frequency of C>T or G>T substitutions was estimated from the strand with the reverse-complementary G>A or C>A substitutions when necessary. Specifically, frequency of $G_{12/13}C$ and $G_{12/13}V$ mutations in *Kras* exon 1 (G>T substitution on the non-transcribed strand) were estimated from the MDS targeting the transcribed strand while frequency of $G_{12/13}S$ and $G_{12/13}D$ mutations in *Kras* exon 1 (C>T substitutions on the transcribed strand) were estimated from MDS targeting the non-transcribed strand.

**Droplet digital PCR (ddPCR)**. ddPCR was performed using the QX200 AutoDG Droplet Digital PCR System (Bio-Rad) following the manufacturer's protocol in a 22 μl ddPCR reaction containing 11 μl of 2x ddPCR SuperMix for probes (no dUTP) (Bio-Rad), 66 ng template DNA, 450 nM forward and reverse primers, and 250 nM FAM- and HEX-labeled probes. The primer and probe oligonucleotides were synthesized (IDT) based on sequences previously described[76] with minor modifications. The sequences for the primers are Kras_Q61_For: 5′-ATGGAGAAACCTGTCTCTTGG-3′ and Kras_Q61_Rev: 5′-CTCATGTACTGGTCCCTCATT-3′. The sequences for the probes are Kras_Q61L_MUT_FAM: 5′-/56-FAM/CAGGT+C+T+AGA+GGAG/3IABkFQ/-3′ and Kras_Q61L_WT_HEX: 5′-/5HEX/CAGGT+C+A+AGA+GGAG/

3IABkFQ/-3′ where "+" denotes the following base is a locked nucleic acid. Following droplet generation on the AutoDG, the plate was sealed with pierceable foil heat seal (Bio-Rad) and PCR performed on a C1000 Touch™ thermal cycler (Bio-Rad). Thermal cycling conditions were as follows: once cycle at 95 °C for 10 min, 40 cycles at 94 °C for 30 s and 60 °C for 60 s, once cycle at 98 °C for 10 min, and 4 °C until the sample was removed. Every ddPCR run included no template control, wild type control with DNA from PBS-treated mice, and mutation-positive control. To achieve detection sensitivity of 1 in 10,000, each sample was assayed in at least two wells. Plates were read on a QX200 droplet reader (Bio-Rad) and analyzed with QuantaSoft™ Analysis Pro software (version 1.0.596) (Bio-Rad) to assess the number of droplets positive for mutant DNA, wild-type DNA, both, or neither. The mutant allele fraction[43] was estimated as follows: The concentration of mutant DNA (copies of mutant DNA per droplet) was estimated from the Poisson distribution using the formula number of mutant copies per droplet $Mmu = -\ln(1-(nmu/n))$, where nmu = number of droplets positive for mutant FAM probe and $n$ = total number of droplets. The DNA concentration in the reaction was estimated using the formula $MDNAconc = -\ln(1-(nDNAcon/n))$, where nDNAconc = number of droplets positive for mutant FAM probe and/or wild-type HEX probe and $n$ = total number of droplets. The mutant allele fraction = Mmu/MDNAconc.

**RNA isolation and quantitative PCR**. RNA was extracted from the lung, liver and pancreas of 6-week-old A/J mice using TRIzol (Thermo Fisher Scientific) and converted to cDNA using iScript™ cDNA Synthesis Kit (Bio-Rad) following the manufacturer's instructions. Quantitative PCR reactions were performed using iTaq Universal SYBR Green Supermix (Bio-Rad) and CFX384 touch real-time PCR detection system (Bio-Rad) using the forward (5′-CCAGCGTCGTGATTAGCGA-3′) and reverse (5′-CCAGCAGGTCAGCAAAGAAC-3′) primers (IDT) to detect the control *Hprt* mRNA and the forward (5′-GCAAGAGCGCCTTGACGATA-3′) and reverse (5′-CATGTACTGGTCCCTCATTGCAC-3′) primers (IDT) to detect *Kras* mRNA. Gene expression values were calculated using the comparative Ct $(-\Delta\Delta Ct)$ method[77], using *Hprt* housekeeping gene as internal control.

**Whole-exome analysis of mutation frequency versus gene expression**. Mutation counts were obtained from published datasets[3] [https://www.nature.com/articles/nature13898#Sec19]. Single-nucleotide variations (SNVs) identified by the whole-exome sequencing of urethane-induced adenomas and adenocarcinomas were examined. The expression level of the genes containing these SNVs were determined from published datasets[36] [https://www.nature.com/articles/s41598-017-04520-z#Sec18]. FPKM values of genes expressed in the lung of 6-weeks-old C57BL/6JJcl mice were used. The second set of gene expression data[37] were obtained from mouse ENCODE project [http://chromosome.sdsc.edu/mouse/download.html]. FPKM values of genes expressed in the lung of 8-week-old male C57Bl/6 mice were used. To bin the genes into different expression groups, the genes were sorted by the mean FPKM value across biological replicates and split into quartiles. The sum of CAN→CTN transversions in the non-transcribed or transcribed strand for the genes in each quartile was calculated and the mean ± SEM of all tumors was plotted.

**Generation of heatmaps**. All heatmaps were generated using Morpheus (https://software.broadinstitute.org/morpheus). All mutation frequencies used in heatmap were corrected by the addition of the detection limit at a barcode recovery of 1.5 × 10⁵ (~6.67 × 10⁻⁶). For the heatmap showing the mutation frequency per nucleotide (Fig. 1b), the sum of the corrected mutation frequencies for all substitutions at an individual nucleotide was obtained, then the fold change of each urethane-treated sample versus the average of PBS-treated samples was calculated and log₁₀ transformed for plotting. For the heatmaps showing the mutation frequency of each type of substitution (Figs. 1c, 2a, and 3a), the sum of the corrected mutation frequencies for each type of substitutions was obtained, then the fold change of each urethane-treated sample versus the average of PBS-treated samples was calculated and log₁₀ transformed for plotting. For the heatmaps showing the mutation frequency of A>T transversions (Figs. 1d, 2b, and 3b; Supplementary Figs. 2a and 3a), the corrected mutation frequency for each A>T transversion was log₁₀ transformed and plotted.

**Statistics**. The number of independent experiments and the statistical analysis used are indicated in the legends of each figure. Data are represented as mean ± SEM. $p$ values were determined by Holm-Sidak multiple comparisons test following one-way or two-way ANOVA, non-parametric Dunn's multiple comparison test following Kruskal–Wallis test, or two-tailed non-parametric Mann–Whitney U test. $p < 0.05$ was considered significant. Different levels of significance are indicated as $*p < 0.05$, $**p < 0.01$, $***p < 0.001$, $****p < 0.0001$ and ns: not significant. Holm-Sidak multiple comparisons test following ANOVA and non-parametric Dunn's multiple comparison test following Kruskal–Wallis test were executed using GraphPad Prism 6. Two-tailed non-parametric Mann–Whitney U test was executed using excel supplemented with Real Statistics Resource Pack (www.real-statistics.com).

**Reporting summary**. Further information on research design is available in the Nature Research Reporting Summary linked to this article.

## Data availability

All raw Illumina® sequencing data has been deposited to NCBI Sequence Read Archive (SRA) under accession number PRJNA561927. The remaining data can be found within the Article, Supplementary Information or available from the authors upon reasonable request.

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

## Acknowledgements

This work was supported by the National Institute of Cancer (R01CA94184 and P01CA203657 to C.M.C.), the National Institute of General Medicine (R35GM127062 to D.M.M.), and a pilot grant from the Duke Cancer Institute and the Nicholas School of the Environment (to C.M.C. and D.M.M.). We thank Dr. Joel Meyer and members of the Counter laboratory for helpful discussions, Dr. Ivan Spasojevic and Ping Fan from Duke Cancer Institute Pharmacokinetics & Investigational Chemotherapy Shared Resource for performing pharmacokinetic analysis of urethane and vinyl carbamate, and colleagues from Duke Cancer Institute Sequencing and Genomic Technologies Shared Resource for performing the quality check and sequencing of NGS libraries.

## Author contributions

S.L. performed the experiments with D.M.M. advising and C.M.C. supervising. All authors helped prepare the paper.

## Competing interests

C.M.C. is a co-founder of Merlon Inc, which played no role in the design or execution of this study. The authors declare no competing interests.
