## [Peer Review File · Nature Communications]

Reviewers' comments:

Reviewer #1 (Remarks to the Author): Expertise in MDS sequencing

In this manuscript, Li et al. explores the appearance of a specific Ras mutation in response to the carcinogen urethane. It is known that different cancers are predisposed to specific mutations in certain codons, proto-oncogene isoforms, and organs, but the reasons for this preference is not fully understood. Since urethane induces a specific Kras Q61L or Q61R mutation in the lung, it is an excellent model system for probing this question.

Li et al. primarily uses maximum depth sequencing (MDS): an error-corrected, high-throughput method for detecting ultra-rare mutations. Conventional next generation sequencing (NGS) is error prone with a lower detection limit of 10^{-3} while MDS allows for detection of extremely rare mutants to $<10^{-6}$. Leveraging this method, they can detect the very first initiating events of urethane mutagenesis. In addition, MDS allows for strand-specific sequencing, illuminating a transcriptional bias in mutation rate that would not be possible with conventional techniques.

Using this method Li et al. make several conclusions. Although it is known that urethane is biased towards A>T and A>G substitutions, they further identify that these substitutions are more likely when a C precedes the A. The authors determine that urethane mutagenesis is not specific to Kras; they arise in the Hras isoform in the lung as well, but only the Kras mutant is pathogenic and accumulates. Furthermore, they find that Kras mutations do not accumulate in the liver and pancreas. This effect is not correlated with urethane availability in those organs, but instead with the transcriptional status of the proto-oncogene.

The research question asked here is novel and the authors provide interesting conclusions based on the sequencing data they collected. However, these conclusions could be explored more thoroughly and, in some cases, need more substantial evidence. The following comments outline suggested improvements to the manuscript:

Major:

1. In several figures (1e,f, 2c, 3c,d, 4a, b) the authors use open circles to denote samples where no mutations are found but then give these samples a non-zero value at the sample's detection limit. If no mutations are detected, the mutation frequency should be zero. Figure 4a shows that there is a significant difference in the frequency of mutations found in liver and pancreas treated with urethane, however each organ is shown to be unaffected by carcinogen when compared to vehicle. This suggests that there is an increased mutation frequency in the liver independent of carcinogen. This is likely an artifact from using the sample's detection limit instead of zero to denote undetected mutations.
2. In figure 3d the authors conclude that Hras acquires mutations at a detectable frequency but do not expand appreciably over time. The data shows there is no significant difference in mutation frequencies at 4 weeks between Kras and Hras treated with carcinogen.
3. The authors propose that increased Kras transcription and thus increased transcription-coupled damage to be responsible for organ tropism of Kras mutations. Conversely, it has also been shown that highly transcribed regions in cancer cells harbor fewer mutations on both strands (PMID: 25456125). The authors might consider addressing this disagreement by revisiting strand specific sequencing in the liver and pancreas where Kras is less highly transcribed, or by observing mutation accumulation in loci with different levels of transcription. This observation warrants further investigation.
4. Confirmation of increased Kras transcription in the lung compared to the liver and pancreas, especially in the context of urethane exposure, would be valuable. Expression levels of Hras in the

lung could also add valuable information.

5. The authors suggest several times that their results “could have clinical implications for early detection and preventative measures” (28). Please expand on this comment as this is not immediately clear.

Minor:

6. There is some literature that shows urethane initially induces adenomas that progress much later to adenocarcinomas, and that secondary mutations in p53 might be involved (PMID: 8989915). It might be more interesting to look at this locus over longer time periods to observe the initiation of secondary mutations.

7. The legend in figure 4 references figure c and d which do not exist.

8. Axes breaks skew data and make interpretation of the distribution and significance of results difficult. It is suggested to log transform the data.

9. Supplementary figure 3c should be moved to figure 2 to demonstrate that the G12 and G13 mutations do indeed arise but at a lower frequency than Q61 mutations.

10. It would be clearer if all figures included information about whether the data is from the non-transcribed, transcribed, or both strands, and which week post-carcinogen. For example, figure 3c includes the week but not the strand, while figure 5a includes neither the week nor the strand.

Reviewer #2 (Remarks to the Author): Expertise in molecular carcinogenesis

RE: NCOMMS-19-30109-T.

Li et al.

Capturing the primordial Kras mutation initiating tumorigenesis.

COMMENTS TO THE AUTHORS

With great interest I reviewed the above-referenced manuscript for Nat Commun. The study aimed at capturing the initiating Kras mutation induced by the tobacco carcinogen urethane in vivo using massive depth sequencing (MDS), already reported to detect rare mutational events in bacteria. The study is elegant and important, as this technique is likely going to provide insights into mutational processes, and as the model employed is highly relevant to human lung adenocarcinoma of smokers. However, several shortcomings hamper the potential impact and appeal of the paper, mainly related to methodology, data presentation, data validation, and sample size, criticisms detailed in the comments below.

MAJOR COMMENTS FOR REVISION

1. Despite the elegant reshape and the pivotal role of MDS for the study, it would be of general interest to have it explained in one single paragraph and not divided among main text and methods as it is in the manuscript.

2. Moreover a validation of the main finding through another technique would be interesting since the single comparison among MDS and NGS is not reliable due to difference in the detection power. For this aim, digital droplet PCR with a million partitions (Sensors (Basel). 2018 Apr 20;18(4). pii: E1271. doi: 10.3390/s18041271) and specific KrasQ61L probes could be adapted for high sensitivity.

3. In line with the above, we performed ddPCR in the urethane model before (in lineage-marked cells) and found slightly higher numbers of mutations (Elife. 2019 May 29;8. pii: e45571. doi: 10.7554/eLife.45571). Capture of the initiating oncogenic mutation in Kras is well addressed and also confirmed from the expansion of the rate during time in the present study. Despite that, the authors should mention and discuss the different findings from the two studies and any other existing studies from the past using traditional techniques (Cancer Lett. 1996 Oct 22;107(2):165-70).

4. One last comment is regarding the sample size. Although the really promising results shown and the several ways of testing the findings are interesting, in the statistics section all the test performed are post data collection. The authors should definitely employ some a priori power analyses and should definitely increase the sample size in some instances to a minimum of five. One to three observations are definitely not enough to achieve an alpha error of less than 0,05.

Reviewer #3 (Remarks to the Author): Expertise in RAS (in vivo)

NCOMMS-19-30109-T. Title: Capturing the primordial Kras mutation initiating tumorigenesis.

Summary and recommendations:

KRAS is the most frequently mutated oncogene in cancer (~10%) and is in particular mutated with high frequency in lung, colorectal and pancreatic tumors. A central question to our understanding of this oncogene is why particular oncogenic alleles of KRAS are found with particular frequencies in different tissues. Li, MacAlpine, and Counter define this aspect of KRAS in cancer as 'tropism' and point out that despite this observation being made decades ago we still do not have an answer. To this end, Li et al. developed an ultrasensitive PCR technique for the detection of particular exon 1 and 2 mutations in KRAS (and other isoforms). Li et al. firmly establish the technique using a carcinogen model to detect enrichment of Q61L mutations in mouse lung, which is completely consistent with the literature involving this mouse model. Their technique has the potential to increase KRAS mutation detection in patients, allowing earlier discrimination of tumors with KRAS mutations and therefore better approaches to individual patient treatment. Furthermore the technique is likely to be useful in research, and their characterization and description of the PCR technique suggests it will be easy to reproduce. However, their studies on lung carcinogenesis using urethane in mouse models, although a great model to test their technique, are artificial and generally provide information of urethane 'tropism' rather than why a particular KRAS mutation (i.e. Q61L) is selected for in lung cancer. In further support, G12 mutations are far more common than Q61 mutations in human lung cancer, and urethane exposure in human carcinogenesis occurs through oral digestion rather than intraperitoneal injection. Therefore I do not recommend 'Capturing the primordial Kras mutation initiating tumorigenesis' by Li et al. for publication in Nature communications (per criteria https://www.nature.com/authors/editorial_policies/peer_review.html) because the data presented here do not enhance our understanding of KRAS biology, in cancer or otherwise, despite the important impact that the PCR technique will have on the field. Inclusion of more biologically or clinically relevant experiments, such as patient tumor sequencing, further investigation of results in Fig. 1B (see below), or a more universal carcinogen such as cigarette smoke or alcohol consumption, would warrant publication at a later time.

More specific comments are listed below.

1. Fig. 1A is not clear.
2. Fig. 1B clearly shows that a number of different alterations are present by week 2. We know that urethane induces lung tumorigenesis with K-Ras Q61 alleles present, but if other alleles are present (Fig. 2C), exploration of other urethane-targeted genes might describe a signaling context necessary for K-Ras Q61L promoted tumorigenesis. This would be interesting considering the

context of senescence that has been demonstrated for K-Ras induction by Q61L.

3. A useful experiment might be to track the kinetics of Q61L mutation and amplification over a longer time frame. Again, it is clear that urethane induces Q61L as shown by others. However, your experiments do not directly show that these very early mutation events necessarily induce tumors. I would anticipate that the density of Q61L mutations might show an inflection representing a shift from hyperproliferation to actual tumor growth. These experiments could be easily paralleled using histology or other techniques.

4. Why are data in Lung-V column split in Fig. 4A? Is this an artifact of data processing?

5. Statistics seem appropriate.

Reviewer 1

Major comment 1: *“In several figures (1e,f, 2c, 3c,d, 4a, b) the authors use open circles to denote samples where no mutations are found but then give these samples a non-zero value at the sample’s detection limit. If no mutations are detected, the mutation frequency should be zero. Figure 4a shows that there is a significant difference in the frequency of mutations found in liver and pancreas treated with urethane, however each organ is shown to be unaffected by carcinogen when compared to vehicle. This suggests that there is an increased mutation frequency in the liver independent of carcinogen. This is likely an artifact from using the sample’s detection limit instead of zero to denote undetected mutations.”*

As requested, we originally chose to use open symbols arbitrarily set above 0 to denote an absence of mutations within a specific detection limit. However, we **completely agree** that this not only artificially skews the data, but is also a very cumbersome and confusing way of representing the findings. We have therefore replotted all data in bar graphs with an absence of mutations assigned a value of 0 (or 10^{-7} when a log scale is used) with the detection limits provided in Supplementary Table 3. This is actually the way the original MDS results were plotted, and in retrospect we should have followed their lead.

Major comment 2: *“In figure 3d the authors conclude that Hras acquires mutations at a detectable frequency but do not expand appreciably over time. The data shows there is no significant difference in mutation frequencies at 4 weeks between Kras and Hras treated with carcinogen.”*

As requested, we note here that the original statistical analysis in Figure 3 was performed on all mutations excluding Q₆₁L to make the point that the mutational load was the same between *Kras* and *Hras*, which is admittedly was not at all clear. We have repeated the statistical comparisons using nonparametric analysis that take into account all mutations. While there is not statistical difference between all mutations, it is very clear from replotting the data that *Kras*^{Q₆₁L} mutations expand (e.g. at 4 weeks they are detected at a frequency of $1-3 \times 10^{-4}$) while *Hras*^{Q₆₁L} mutations do not (e.g. at 4 weeks they are detected at a frequency of $1-2 \times 10^{-5}$, which is similar to all non-oncogenic mutations). Thank you for pointing this out.

Major comment 3: *“The authors propose that increased Kras transcription and thus increased transcription-coupled damage to be responsible for organ tropism of Kras mutations. Conversely, it has also been shown that highly transcribed regions in cancer cells harbor fewer mutations on both strands (PMID: 25456125). The authors might consider addressing this disagreement by revisiting strand specific sequencing in the liver and pancreas where Kras is less highly transcribed, or by observing mutation accumulation in loci with different levels of transcription. This observation warrants further investigation.”*

As requested, we now demonstrate that there is a decrease in expression (Fig 5c), a significant drop in mutations (Fig 4a), and a loss of strand bias (Fig 5d) in *Kras* gene of the liver compared to the lung. We also attempted to test another locus in the same fashion, but were unsuccessful, and have shied away from cherry-picking genes to test given the labor intensive

nature of developing MDS for a new locus. Instead, we compared the rate of transcription to the mutation frequency across the coding genome of tumors derived from urethane exposure (Fig 5e), finding strand-specific correlation between increased gene expression and mutation frequency. Lastly, we discuss the cited work in the revised text.

Major comment 4: *“Confirmation of increased Kras transcription in the lung compared to the liver and pancreas, especially in the context of urethane exposure, would be valuable. Expression levels of Hras in the lung could also add valuable information.”*

As requested, we performed the suggested RT-qPCR analysis of *Kras* mRNA and confirm the published observation of that this transcript is more highly expressed in the lung compared to the liver or pancreas (Fig 5c). Given this concordance with published datasets, we took advantage of previous quantification of Ras isoforms in the lung to directly compare levels of different isoforms. We note here that *Hras* is either lower (PMID: 28117393) or comparable to *Kras* (PMID: 22763441; 28646208) mRNA levels, consistent with our finding that *Hras* is mutated at an appreciable frequency in the lung, but these oncogenic alleles do not expand over time.

Major comment 5: *“The authors suggest several times that their results “could have clinical implications for early detection and preventative measures” (28). Please expand on this comment as this is not immediately clear.”*

As requested, we removed this statement.

Minor comment 1: *“There is some literature that shows urethane initially induces adenomas that progress much later to adenocarcinomas, and that secondary mutations in p53 might be involved (PMID: 8989915). It might be more interesting to look at this locus over longer time periods to observe the initiation of secondary mutations.”*

We absolutely agree that expanding this study to explore when secondary mutations arise is a great line of future investigation. The challenge is that it takes about 25 weeks for adenomas and 40 weeks for adenocarcinomas to develop (PMID: 26330618), with the latter accounting for less than 10% of the tumors at 56 weeks (PMID: 3470549). Nevertheless, this is an excellent suggestion, and something we definitely plan on pursuing in the future. Great idea.

Minor comment 2: *“The legend in figure 4 references figure c and d which do not exist.”*

As requested, we corrected this mistake, thank you for catching it.

Minor comment 3: *“Axes breaks skew data and make interpretation of the distribution and significance of results difficult. It is suggested to log transform the data.”*

As requested, we replotted Figures 1e, 2d, 3d, and 4b using a log scale.

Minor comment 4: *“Supplementary figure 3c should be moved to figure 2 to demonstrate that the G12 and G13 mutations do indeed arise but at a lower frequency than Q61 mutations.”*

As requested, we moved supplementary Figure 3c to Figure 2.

Minor comment 5: *“It would be clearer if all figures included information about whether the data is from the non-transcribed, transcribed, or both strands, and which week post-carcinogen. For example, figure 3c includes the week but not the strand, while figure 5a includes neither the week nor the strand.”*

As requested, we added the week after urethane exposure to all graphs and added the strand that was sequenced to all the figure legends, except in Figure 5, where the strand sequenced was denoted in the figure for comparative purposes. If the reviewer wishes that the strand information also be indicated in the panels we can do that as well.

Reviewer 2

Major comment 1: *“Despite the elegant reshape and the pivotal role of MDS for the study, it would be of general interest to have it explained in one single paragraph and not divided among main text and methods as it is in the manuscript.”*

As requested, we have revised the text so that MDS assay is explained in detail in one paragraph.

Major comment 2: *“Moreover a validation of the main finding through another technique would be interesting since the single comparison among MDS and NGS is not reliable due to difference in the detection power. For this aim, digital droplet PCR with a million partitions (Sensors (Basel). 2018 Apr 20;18(4). pii: E1271. doi: 10.3390/s18041271) and specific KrasQ61L probes could be adapted for high sensitivity.”*

As requested, we validated the mutation frequency of the Q₆₁L mutation in mouse lung 4 weeks after urethane exposure using digital droplet PCR (Supplementary Fig 1d).

Major comment 3: *“In line with the above, we performed ddPCR in the urethane model before (in lineage-marked cells) and found slightly higher numbers of mutations (Elife. 2019 May 29;8. pii: e45571. doi: 10.7554/eLife.45571). Capture of the initiating oncogenic mutation in Kras is well addressed and also confirmed from the expansion of the rate during time in the present study. Despite that, the authors should mention and discuss the different findings from the two studies and any other existing studies from the past using traditional techniques (Cancer Lett. 1996 Oct 22;107(2):165-70).”*

As requested, we apologize for our oversight in not including these insightful papers and now discuss these studies in the revised text.

Major comment 4: *“One last comment is regarding the sample size. Although the really promising results shown and the several ways of testing the findings are interesting, in the statistics section all the test performed are post data collection. The authors should definitely employ some a priori power analyses and should definitely increase the sample size in some instances to a minimum of five. One to three observations are definitely not enough to achieve an alpha error of less than 0,05.”*

As requested, we note here that we performed a priori power analysis using G*Power and obtained a sample size of 4, assuming $\alpha = 0.05$, $\beta = 0.05$, and effect size $d = 3.2$. The effect size was calculated using G*Power from the comparison of the mutation frequency of less common oncogenic Q₆₁ mutations detected (A₁₈₂G (Q₆₁R) and A₁₈₃C (Q₆₁H)) in Kras exon 2 one week after urethane exposure. We thus repeated the entire experiment to increase sample sizes to a minimum of 4 mice in almost all cohorts.

Reviewer 3

Major comment 1: *“...their studies on lung carcinogenesis using urethane in mouse models, although a great model to test their technique, are artificial and generally provide information of urethane ‘tropism’ rather than why a particular KRAS mutation (i.e. Q61L) is selected for in lung cancer.”*

As requested, we made major revisions to the text, from the title to the discussion, to focus on the RAS mutation tropism of urethane. We only discuss the relevance to human RAS mutation patterns in the discussion. We heard you loud and clear and completely repositioned this manuscript as requested.

Major comment 2: *“...G12 mutations are far more common than Q61 mutations in human lung cancer”*

We agree, and in fact this study demonstrates that this is in large part because urethane does fails to generate G₁₂ mutations at a high frequency. As noted above, we restructured the manuscript to focus on urethane carcinogenesis, and further, include the caveat that urethane does not induce the same mutations as typically seen in human lung cancer.

Major comment 3: *“...urethane exposure in human carcinogenesis occurs through oral digestion rather than intraperitoneal injection.”*

In reply, we note here that the advantage of intraperitoneal injection is that *i)* urethane has the opportunity to penetrate many tissues, which allowed us to study the tissue tropism of this carcinogen, *ii)* the dose can be tightly controlled, and *iii)* it is the most studied route (PMID: 2654935, PMID: 8989915, PMID: 17205523, PMID: 25363767), which allowed us to capitalize upon decades of research, which was critical to Figure 5e.

Major comment 4: *“...inclusion of more biologically or clinically relevant experiments, such as patient tumor sequencing...”*

In reply, we agree that human tissues is certainly the most relevant, but it is not possible to capture an initiating event after carcinogen exposure in humans, and then follow the outcome, which was the experimental design of this manuscript. However, we see this reviewer’s point and discuss the results of our study in comparison to the type of RAS mutations detected in by sequencing human cancers.

Major comment 5: *“...[study] a more universal carcinogen such as cigarette smoke or alcohol consumption...”*

We agree that expanding this study to other carcinogens will be an important line of future investigation, but we first wanted to examine how the most studied of these carcinogens in terms of RAS mutation tropism behaved as a starting point. Nevertheless, we indicate the caveat that urethane is not a common carcinogen in humans compared to cigarette smoke.

Minor comment 1: *“Fig. 1A is not clear.”*

As requested, we simplified and relabeled this panel to make it clearer.

Minor comment 2: *“Fig. 1B clearly shows that a number of different alterations are present by week 2. We know that urethane induces lung tumorigenesis with K-Ras Q61 alleles present, but if other alleles are present (Fig. 2C), exploration of other urethane-targeted genes might describe a signaling context necessary for K-Ras Q61L promoted tumorigenesis. This would be interesting considering the context of senescence that has been demonstrated for K-Ras induction by Q61L.”*

In reply, this is a great point, and one that has been explored by whole-exome sequencing of urethane-induced tumors (PMID: 25363767), with the finding there is no evidence for any one specific cooperating mutation in urethane carcinogenesis, although this does not formally exclude the possibility of a wide spectrum of cooperating mutations being able to nudge a *Kras* mutation down the path of tumorigenesis. Please also note that MDS cannot be used to screen for co-occurring mutations (unless there was a specific one to sequence).

Minor comment 3: *“A useful experiment might be to track the kinetics of Q61L mutation and amplification over a longer time frame. Again, it is clear that urethane induces Q61L as shown by others. However, your experiments do not directly show that these very early mutation events necessarily induce tumors. I would anticipate that the density of Q61L mutations might show an inflection representing a shift from hyperproliferation to actual tumor growth. These experiments could be easily paralleled using histology or other techniques.”*

In reply, we note here that we detect Q₆₁L mutations at one week, they dramatically increase over time up to the last time point of one month (some 50-fold over non-oncogenic mutations), and exhaustive analysis by multiple groups has shown that the resulting tumors are *Kras*^{Q61L}-mutant (PMID: 2654935, PMID: 8989915, PMID: 19609923), consistent with these being initiating events. As a technical point, only about 10% of urethane-induced tumors (PMID: 3470549) progress to adenocarcinomas (indicative of cooperating mutations), a process that takes 40 weeks (PMID: 26330618), making the suggested experiment a challenge. Nevertheless, we note the caveat that a longer time course would validate the initiating nature of these mutations.

Minor comment 4: *“Why are data in Lung-V column split in Fig. 4A? Is this an artifact of data processing?”*

As requested, we note that this was due to the way we presented the data to reflect sequencing depth. However, reviewer 1 asked us to provide the data with the absence of mutations assigned a value of 0, which will greatly clarify the results and removes the artificial splitting of the data.

REVIEWERS' COMMENTS:

Reviewer #1 (Remarks to the Author):

The authors have provided satisfactory responses to each of the major points raised:

1. The figures have been improved and made more intuitive with this change.
2. Good to see this resolved.
3. The additional figure showing mutation rate and strand bias in liver Kras addresses this point. However, the discussion of citation 63 on line 315 is incorrect. The key finding of Zheng et al., 2014 is that transcription is associated with lower mutation rates in chromatin dense regions, which is the opposite of the findings in this manuscript under review. Future work could investigate why this discrepancy exists.
4. This comment is addressed.
5. The text has been revised significantly and now represents the data in a more legitimate way.

Reviewer #2 (Remarks to the Author):

The authors addressed all this reviewer's concerns.

Reviewer #3 (Remarks to the Author):

Li, MacAlpine, and Counter have addressed all of my comments and concerns related to their manuscript titled: Capturing the primordial Kras mutation initiating urethane carcinogenesis. I believe the technique described will be very useful as the field continues to focus on the role of KRAS alleles in oncogenesis, and will be applicable to other research areas as well.

One minor issue that I did note on my second reading that if addressed would certainly improve the readability is the use of clear units when describing the efficacy of MDS versus NGS (e.g. lines 33, 39, 50 etc.).

COMMENTS OF REVIEWER #1

39) *“However, the discussion of citation 63 on line 315 is incorrect. The key finding of Zheng et al., 2014 is that transcription is associated with lower mutation rates in chromatin dense regions, which is the opposite of the findings in this manuscript under review. Future work could investigate why this discrepancy exists.”*

As requested, we corrected the statement of the key finding of citation 63 as suggested.

COMMENTS OF REVIEWER #2

None to address.

COMMENTS OF REVIEWER #3

40) *“One minor issue that I did note on my second reading that if addressed would certainly improve the readability is the use of clear units when describing the efficacy of MDS versus NGS (e.g. lines 33, 39, 50 etc).”*

As requested, we now include the units as recommended.